A fresh look at Cladarosymblema narrienense, a tetrapodomorph fish (Sarcopterygii: Megalichthyidae) from the Carboniferous of Australia, illuminated via X-ray tomography

Clement Alice M. alice.clement@flinders.edu.au 1
Cloutier Richard 2
Lu Jing 3 4
Perilli Egon 1
Maksimenko Anton 5
Long John 1
1 College of Science and Engineering, Flinders University , Adelaide , South Australia , Australia
2 Département de Biologie, Chimie et Géographie, University of Québec at Rimouski , Rimouski , Quebec , Canada
3 Key Laboratory of Vertebrate Evolution and Human Origins of Chinese Academy of Sciences, Institute of Vertebrate Paleontology and Paleoanthropology , Beijing , China
4 CAS Center for Excellence in Life and Paleoenvironment , Beijing , China
5 Australian Synchrotron, Australian Nuclear Science and Technology Organisation , Melbourne , Victoria , Australia
Hutchinson John
Electronic publication date: 2021 Dec 10
Publication date: 2021
Volume: 9
Electronic Location ID: e12597
Received 2021 Sep 28; Accepted 2021 Nov 14
Copyright: ©2021 Clement et al.
Copyright year: 2021
Copyright holder: Clement et al.
License: This is an open access article distributed under the terms of the Creative Commons Attribution License, which permits unrestricted use, distribution, reproduction and adaptation in any medium and for any purpose provided that it is properly attributed. For attribution, the original author(s), title, publication source (PeerJ) and either DOI or URL of the article must be cited.
License URL: https://creativecommons.org/licenses/by/4.0/

Keywords: Sarcopterygii, Tetrapodomorph, Megalichthyidae, Carboniferous, Tomography, Endocast, Phylogenetic analysis, 3D modelling, Evolution, Vertebrate

Funding: The Australian Research Council DP160102460 LE180100136 DP200103398 Flinders University Impact Seed Funding an NSERC Discovery Grant The Strategic Priority Research Program of Chinese Academy of Sciences XDB26000000 National Science Fund for Excellent Young Scholars 42022011 National Natural Science Foundation of China 41872023 This work was supported by the Australian Research Council (DP160102460, LE180100136, DP200103398), Flinders University Impact Seed Funding (to Alice M Clement), an NSERC Discovery Grant (to Richard Cloutier), and the following all awarded to Jing Lu: the Strategic Priority Research Program of Chinese Academy of Sciences (XDB26000000), National Science Fund for Excellent Young Scholars (42022011), and National Natural Science Foundation of China (41872023). The funders had no role in study design, data collection and analysis, decision to publish, or preparation of the manuscript.

==============================
Background

The megalichthyids are one of several clades of extinct tetrapodomorph fish that lived throughout the Devonian–Permian periods. They are advanced “osteolepidid-grade” fishes that lived in freshwater swamp and lake environments, with some taxa growing to very large sizes. They bear cosmine-covered bones and a large premaxillary tusk that lies lingually to a row of small teeth. Diagnosis of the family remains controversial with various authors revising it several times in recent works. There are fewer than 10 genera known globally, and only one member definitively identified from Gondwana. Cladarosymblema narrienense Fox et al. 1995 was described from the Lower Carboniferous Raymond Formation in Queensland, Australia, on the basis of several well-preserved specimens. Despite this detailed work, several aspects of its anatomy remain undescribed.

Methods

Two especially well-preserved 3D fossils of Cladarosymblema narrienense, including the holotype specimen, are scanned using synchrotron or micro-computed tomography (µCT), and 3D modelled using specialist segmentation and visualisation software. New anatomical detail, in particular internal anatomy, is revealed for the first time in this taxon. A novel phylogenetic matrix, adapted from other recent work on tetrapodomorphs, is used to clarify the interrelationships of the megalichthyids and confirm the phylogenetic position of C. narrienense.

Results

Never before seen morphological details of the palate, hyoid arch, basibranchial skeleton, pectoral girdle and axial skeleton are revealed and described. Several additional features are confirmed or updated from the original description. Moreover, the first full, virtual cranial endocast of any tetrapodomorph fish is presented and described, giving insight into the early neural adaptations in this group. Phylogenetic analysis confirms the monophyly of the Megalichthyidae with seven genera included (Askerichthys, Cladarosymblema, Ectosteorhachis, Mahalalepis, Megalichthys, Palatinichthys, and Sengoerichthys). The position of the megalichthyids as sister group to canowindrids, crownward of “osteolepidids” (e.g.,Osteolepis and Gogonasus), but below “tristichopterids” such as Eusthenopteron is confirmed, but our findings suggest further work is required to resolve megalichthyid interrelationships.

Introduction

Megalichthyids are an extinct clade of sarcopterygian (lobe-finned) tetrapodomorph fishes known from predominantly freshwater deposits in the Palaeozoic. They appeared in the Mid-Late Devonian, and were one of the few sarcopterygian groups that survived the end-Devonian extinctions, persisting up until the Lower Permian (Witzmann & Schoch, 2012). They fall within an “osteolepidid-grade” in most phylogenetic analyses of stem-tetrapod interrelationships (Ahlberg & Johanson, 1998; Cloutier et al., 2020; Johanson, 2004; Johanson & Ahlberg, 2001; Lu et al., 2012; Zhu & Ahlberg, 2004; Zhu et al., 2017). They are typically recovered most closely related to the East Gondwanan endemic group the Canowindridae (Beelarongia, Koharalepis, Canowindra), usually crownward of rhizodonts and basal of the tristichopterids (such as Eusthenopteron) and the elpistostegalid fishes.

It was Smith-Woodward (1891) who first grouped Megalichthys, Osteolepis, Thursius, Diplopterus (now Heddleichthys) and Glyptopomus in the family Osteolepidae (correctly the Osteolepididae). Within a paraphyletic “Osteolepididae”, Cloutier & Ahlberg (1996) mentioned that megalichthyids could be recognized as a clade based on several cranial characters referring to Young, Long & Ritchie (1992). Since then, “osteolepiforms” have been unquestionably recognized to be paraphyletic with respect to elpistostegalians and tetrapods (Ahlberg & Johanson, 1998; Cloutier et al., 2020) and the phylogenetic position and status of the Megalichthyidae have continued to be discussed. Fox et al. (1995, p. 106) considered the Megalichthyidae to be closer to the “Osteolepididae” than to any other families of “Osteolepiformes”. A paraphyletic “Osteolepididae” including a monophyletic group of megalichthyids was also recovered by Ahlberg & Johanson (1998).

More specifically, Hay (1902) was the first to coin the term “Megalichthyidae”, after which Long (1985) suggested synapomorphies to define this particular “osteolepidid” clade, but the first full familial description was not provided until Young, Long & Ritchie (1992). This was later revised by Fox et al. (1995), Borgen & Nakrem (2016), and again most recently by Downs & Daeschler (2020).

In describing a new species of Megalichthys (M. mullisoni) from the Famennian of USA, Downs & Daeschler (2020) reduced the characters defining the family to three specialised features: (premaxillary tusk that interrupts or lies lingual to the premaxillary marginal tooth row; contact between the subopercular and second [posterior-most] submandibular; and a distinct supratemporal bone), in combination with one plesiomorphic character (cosmine cover on dermal bones).

The family Megalichthyidae includes several taxa from Europe, Russia, Middle East, and North America, but there is only one taxon described from Australia. Cladarosymblema narrienense (Fox et al., 1995) is known from the Lower Carboniferous (Viséan) Raymond Formation in Queensland, Australia, and is the only megalichthyid described from the Southern Hemisphere (Long, Clement & Choo, 2018).

Aside from Cladarosymblema, there are several other genera commonly recognised within the Megalichthyidae: Megalichthys (Agassiz, 1835) contains several species found in Devonian-Carboniferous deposits across North America (Cope, 1882; Downs & Daeschler, 2020), Morocco (Janvier & Martin, 1979), and the UK (Thomson, 1964); Ectosteorhachis nitidus (Cope, 1882) is known from the Lower Permian of the USA (Thomson, 1964); Sengoerichthys ottoman—considered by some as the earliest megalichthyid—from the Frasnian of Turkey (Janvier, Clément & Cloutier, 2007); Palatinichthys laticeps described from the Lower Permian of Germany (Witzmann & Schoch, 2012); and the most recently described megalichthyid genus, Askerichthys heintzi, comes from the Late Carboniferous of Norway (Borgen & Nakrem, 2016). However, Downs & Daeschler (2020) considered that the “unusual combinations of characters” in S. ottoman and P. laticeps precluded them from being megalichthyids.

Several other taxa have been considered at times to share affinities with the megalichthyids, but are usually excluded from most analyses due to lacking diagnostic family features or by being too poorly known. Namely, Fox et al. (1995), Janvier, Clément & Cloutier (2007) and Witzmann & Schoch (2012) excluded the lesser-known genera such as the Permo-Pennsylvanian Lohsania from USA (Sumida, Scott & Wideman, 2005; Thomson & Vaughn, 1968), Megistolepis and Megapomus from the Devonian of Russia (Vorobyeva, 1977), Cryptolepis from the Devonian of Latvia (Lebedev, 1995; Vorobyeva, 1975), and Mahalalepis (Young, Long & Ritchie, 1992) from their studies. The Middle-Late Devonian Mahalalepis resima, from Mount Crean in Antarctica, was named from a single fronto-ethmoidal shield and considered by Young, Long & Ritchie (1992) to be a megalichthyid. If accepted as a megalichthyid, it would represent the oldest member of the clade. Additional material is currently under description and will likely soon more conclusively clarify its taxonomic affinities (J Lu, 2021, pers. comm).

In contrast to some of the taxa named above, C. narrienense is well known, described on the basis of several 3D-preserved specimens exposed by acid-etching and mechanical preparation. Fox et al. (1995) described in detail many aspects of its anatomy - including the dermal skull bones, braincase, mandible, pectoral girdle and fin, limited elements of the axial skeleton and hyoid arch, as well as the teeth and scales. However, the accessibility of modern scanning techniques today now permits a detailed re-examination of C. narrienense to illuminate features of its morphology that remained elusive. Herein we use high-resolution micro-CT (µCT) and synchrotron tomography to reveal unseen features not previously described, including elements of the hyoid arch, palatal bones, axial skeleton and a cranial endocast. In doing so several aspects of its morphology are uncovered that prove useful for supporting a more robust clade of the Megalichthyidae, and provide broader resolution in phylogenetic analyses of this problematic “osteolepidid-grade” of Palaeozoic tetrapodomorph fishes.

Material & Methods

Material

Two exceptionally preserved specimens of the megalichthyid, Cladarosymblema narrienense, from the Lower Carboniferous Raymond Formation of the Officer Basin, Queensland, Australia, were scanned using a cabinet micro-CT system or synchrotron tomography to reveal new internal anatomical detail (Fig. S1).

Figure 1 Micro-CT 3D rendering (116 µm pixel size) of dermal skull and braincase of Cladarosymblema narrienense (QMF 21082).

(A) Skull in dorsal view showing placement of bones on holotype; (B) braincase in ventral view; dermal skull bones, braincase and palatal bones in (C) dorsal and (D) ventral view; (E) skull and cheek in right lateral view; (F) cheek bones in mesial view.

The holotype, housed in the Queensland Museum (QMF 21082), is preserved in a single block of silty limestone and contains the skull, anterior trunk region and both pectoral fins, and was described in detail by Fox et al. (1995). Several bones from the left side of the specimen were removed during preparation at that time so not all paired elements are present on both sides of the specimen, and consequently the right side is the more complete. There are reinforcing metal wires surrounding the perimeter of both pectoral fins.

The second specimen (∼50 mm in length, ∼40 mm in width) is an isolated ethmosphenoid from the Queensland Museum collection (QMF 21083). This specimen has been acid-prepared, and retains small sections of in-filled limestone within the cranial cavity. There is some slight compression of delicate elements internally but the nasal capsules and hypophyseal region of the endocranium are well preserved.

Scanning & segmentation

The holotype (QMF 21082), although relatively large (dimensions approximately 230 mm in length, 200 mm in width, 35 mm in height), was able to be scanned in 2020 at the Flinders University micro-CT Laboratory using a large-volume micro-CT system (Nikon XTH225 ST, Nikon Metrology Tring, Hertfordshire, UK). The specimen was placed in a polystyrene-foam box (transparent to X-rays), with the long specimen axis vertically aligned with the rotation axis of the micro-CT rotation stage. The specimen was scanned using the following parameters: 160 kV; 282 µA (45W), 0.25 mm tin filter, 2.83 s exposure, rotation step 0.1° over 360°; with a resulting voxel size of 58 µm (4056 × 4056 pixel detector), containing the entire specimen in the field of view (SI-Fig. 1E). The tomographic cross-sections were reconstructed using a filtered back-projection algorithm (Nikon CT Pro 3D software) and saved as 8-bit bitmap format images. A stack of up to 4,000 consecutive cross sections was reconstructed, resulting in a height of up to 232 mm. Each cross-section was 3000 × 1100 pixels (corresponding to 174 × 63.8 mm) in size. Images were later subsampled by a factor two to a voxel size of 116 µm (Perilli, Parkinson & Reynolds, 2012).

An isolated ethmosphenoid (QMF 21083) was scanned in 2016 at the Australian Nuclear Science and Technology Organisation (ANSTO) Australian Synchrotron in Melbourne (experiment number AM10403) using the following parameters: 50 kV, 181° deg., 1810 projections; angle step 0.1°; exposure time 0.22 s; object to detector 35 mm; with a resultant voxel size of 12.2 µm.

Reconstructed scan data and associated derived files are deposited on MorphoSource at: https://www.morphosource.org/projects/000383372?utf8=, (or see Fig. S1 for example reconstructed tomographic slice data). Data were segmented manually using thresholding and rendered in MIMICS v.18 & v.19 (Materialise, Leuven, Belgium; http://biomedical.materialise.com/mimics), with Stereolithographs (STL 3D models) of each element exported and available on MorphoSource (https://www.morphosource.org/projects/000383372?utf8=). Animations were compiled in Adobe Premier Pro.

Phylogenetic analysis

The phylogenetic position of the megalichthyids among tetrapodomorphs, as well as the interrelationships among megalichthyids are investigated using the tetrapodomorph character matrix of Cloutier et al. (2020). Cloutier et al. (2020) phylogenetic data matrix (comprising 202 characters coded for 43 taxa) was based on 169 characters from Zhu et al. (2017), 13 characters from Daeschler, Shubin & Jenkins (2006), 11 characters from Swartz (2012) and 5 characters from Cloutier et al. (2020). The new matrix includes 49 taxa.

To the original matrix we included five megalichthyids or taxa assigned to be likely megalichthyids: Sengoerichthys ottoman (Janvier, Clément & Cloutier, 2007), Palatinichthys laticeps (Witzmann & Schoch, 2012), Askerichthys heintzi (Borgen & Nakrem, 2016), Mahalalepis resima (Young, Long & Ritchie, 1992), and Megalichthys mullisoni (Downs & Daeschler, 2020). In addition, we recoded Cladarosymblema based on our new morphological description presented herein, with additional codings for Mahalalepis based on work currently in preparation (J Lu, 2021, pers. comm.) Rather than entering Megalichthys as a composite taxon, we coded M. hibberti and M. laticeps in addition to M. mullisoni all as separate species.

Four new characters are added to Cloutier et al.‘s (2020) matrix: characters 203 [Long medioventral process of premaxilla: (0) absent; (1) present]; 204 [Relative size of anterior tectal and lateral rostral: (0) lateral rostral much larger than anterior tectal, (1) lateral rostral and anterior tectal equidimensional, (2) lateral rostral smaller than anterior tectal]; 205 [Size of bones forming the external nares: (0) anterior tectal and lateral rostral similar in size to the posterior tectal, (1) anterior tectal and lateral rostral much smaller than the posterior tectal]; and lastly 206 [Anterolateral process of supratemporal: (0) absent, (1) present].

Simões & Pierce (2021) changed two codings from Cloutier et al.‘s (2020) matrix: character 62 (anteromedial process of vomer) for Acanthostega and character 106 (opercular process of hyomandibula) for Panderichthys. We agree solely with the modification suggested for the coding of Acanthostega for character 62.

Thus, we performed parsimony analyses on our matrix (http://morphobank.org/permalink/?P3818), which was coded for 49 taxa (including 5 outgroups) and 206 characters using a heuristic search. Additional comments on codings are contained within the matrix on Morphobank. The tree was rooted on a constrained monophyletic outgroup composed of Youngolepis, Diabolepis, Powichthys, Porolepis, and Glyptolepis. A total of 18 multi-state characters were run ordered; in addition to the morphocline defined in Cloutier et al. (2020, Supplementary Information) character 204 was considered as a morphocline. Strict, Adams and 50% majority consensus trees were computed. All analyses were performed in PAUP*v4.0a.

Description

The Dermal Skull

On the skull roof, the course of the lateral line canal can be confirmed as lyre-shaped (Figs. 1A, 1C). It lies close to associated pore group clusters, which are considered likely electroreceptors following the work by King, Hu & Long (2018). A network of tubuli in the snout region provides further evidence that Cladarosymblema narrienense likely had some electroreceptive ability. A single bone posterior and disarticulated from the skull roof is confirmed as the median extrascapular (Fig. 1A) due to the presence of the mesial otic sensory-line canal running through it, with no junction for the main otic sensory canal as would be expected in the lateral bones. The cheek bones were described by Fox et al. (1995) and follow a standard basal tetrapodomorph arrangement (Figs. 1A, 1E, 1F). Those on the holotype are imperfectly preserved so will not be considered further.

Palate

Fox et al. (1995) admitted that the palatal bones of Cladarosymblema narrienense were not well known. Those authors pieced together a composite reconstruction of the parasphenoid from several specimens, but failed to recover complete pterygoid bones. The palatoquadrate complex is preserved within the holotype and revealed in the scan data (Figs. 1C, 1D). It is thin and forms a shallow concavity dorsally. Its margins flex upwards where it abuts the lateral sphenoid portion of the braincase (basipterygoid), the anterolateral face of the oticoccipital and the quadrate articulation posteriorly. In contrast, the posterolateral corner bears a more downturned, smoother edge. Anteriorly the pterygoids are generally flat in the dorsoventral plane, but their posterior half is flexed more into the sagittal plane. In ventral view the right pterygoid bears a longitudinal ridge separating its medial and lateral portions (Fig. 1D).

Submandibular bones

Concerning the gulars and submandibular bones (Figs. 2A, 2E, 2F), Fox et al. (1995) described the lateral (principal) gular of the holotype to have a width of 40% of its total length, but its width is closer to 35%. There was little to no overlap area in its posteromedial corner for the other gular bone. The median gular is about 30% of the length of the lateral gulars, thus slightly smaller than the condition in Askerichthys heintzi (c.40%). It is worthy to note that both Borgen & Nakrem (2016) and Witzmann & Schoch (2012) numbered the submandibular bones posterior to anterior, whereas Fox et al. (1995) numbered them anterior to posterior. Consequently, this impacts identification of “last” (or second last) submandibular bones that may or may not be overlapped by the subopercular. We choose to follow the convention in numbering these bones from anterior to posterior, whereby the “last” submandibular is the most posterior one. There are in fact seven submandibulars in the holotype (Figs. 2A, 2E, 2F), as in Megalichthys sp. and Ectosteorhachis nitidus, although the first one is broken in half. This contrasts with Palatinichthys laticeps which has three submandibulars (Witzmann & Schoch, 2012), and A. heintzi which has six (Borgen & Nakrem, 2016). There is an area for overlap with the subopercular on the last (posterior-most) submandibular only, in contrast to M. mullisoni which is said to have contact between the subopercular and the second [last] submandibular (Downs & Daeschler, 2020). There is no evidence of a groove on the underside of the operculum as seen on QMF21105 (Fox et al., 1995).

Figure 2 Micro-CT 3D rendering (116 µm pixel size) of mandible and submandibular bones of Cladarosymblema narrienense (QMF 21082).

(A) mandibular bones in ventral view showing placement of bones on holotype; mandible in (B) dorsal; (C) lingual; and (D) labial view. (E-F), gulars and submandibular bones shown in isolation, in ventral and dorsal view.

Mandible

Fox et al. (1995) were unable to confidently recognise sutures between dermal bones on the mandible on Cladarosymblema narrienense, despite attempting this by using radiographic imaging. The holotype fails to illuminate this further as much of the dermal bone has eroded and the mandible bears a thick cosmine cover (Figs. 2B–2D). These boundaries were also noted to be difficult to ascertain in Askerichthys heintzi (Borgen & Nakrem, 2016) so this may be a feature common among megalichthyids. As discussed in the original description by Fox et al. (1995), it can be confirmed that there is no oral branch of the mandibular canal in the surangular and that the anterior mandibular fossa received the vomerine tusk. The anterior coronoid is longer than the following two, and there are three tusks present (Figs. 2B–2D). A. heintzi (and also probably Ectosteorhachis nitiuds) are known to possess just two (Borgen & Nakrem, 2016; Thomson, 1964). The parasymphysial tooth plate is known to vary in shape between specimens, that in the holotype forms an elongate triangle covered in small denticles (Figs. 2B, 2C).

Endocranium and endocast

The new data generated permit the reconstruction of the virtual cranial endocast for this taxon. Comparisons are made principally with other tetrapodomorphs for which a full endocast is known, namely Gogonasus andrewsae (Holland 2014), Ectosteorhachis and Megalichthys (Romer, 1937), and Eusthenopteron foordi Jarvik 1955 (Stensiö, 1963). Other Palaeozoic sarcopterygians with complete endocasts depicted in the literature include the dipnomorph Youngolepis praecursor (Chang, 1982), the onychodont Qingmenodus jandemarrai (Lu et al., 2016), the coelacanth Diplocercides kayseri (Stensiö, 1963), several lungfish taxa (Challands, 2015; Clement & Ahlberg, 2014; Clement et al., 2016; Henderson & Challands, 2018; Miles, 1977; Säve-Söderbergh, 1952) and the aïstopod Lethiscus stocki (Pardo et al., 2017). Furthermore, partial yet still informative endocasts are known from the stem-tetrapod Tungsenia paradoxa (Lu et al., 2012), porolepiforms Powichthys thorsteinssoni (Clément & Ahlberg, 2010) and Glyptolepis groenlandica (Stensiö, 1963), the tetrapodomorph Spodichthys buetleri (Snitting, 2008) and early tetrapod genus, Ichthyostega (Clack et al., 2003).

The endocast of the holotype (QMF 21082) measures just over 50 mm long from the base of the olfactory tracts to the vagus nerve (n.X), and 30 mm at its widest point across the labyrinths (Figs. 3A, 3B). As the holotype has suffered some dorsoventral compression during preservation, fine details such as the morphology of the semicircular canals have been lost. Despite this, the gross morphology of the endocast can for the first time be revealed in Cladarosymblema narrienense. The isolated ethmosphenoid (QMF 21083) is well-preserved and has only a little localised crushing internally, and so revealed the olfactory and hypophyseal regions particularly well (Figs. 4D–4H).

Figure 3 Micro-CT 3D (116 µm pixel size) and synchrotron rendering (12 µm pixel size) of cranial endocast and sensory lines of Cladarosymblema narrienense (QMF 21082/3).

(A) dorsal; (B) ventral; and (C) left lateral view; QMF 21083 in (D,E) dorsal view; (F,G) ventral view; (H) left lateral view showing zoomed in hypophysial fossa region.

Figure 4 Micro-CT 3D rendering (116 µm pixel size) of hyoid and branchial skeleton of Cladarosymblema narrienense (QMF 21082).

(A) skull in dorsal view showing placement of bones on holotype; (B) in ventral view; (C,D) full hyoid and basibranchial skeleton including ceratobranchials as preserved in situ; (E) closeup of left hyomandibular; and right hypobranchials 1-4 in (F) ventral and (G) dorsal view.

The overall proportions of the endocast in C. narrienense are similar to those in Y. praecursor, G. andrewsae, Megalichthys spp., and Eusthenopteron foordi in having widely separated nasal capsules on long olfactory tracts, a narrow forebrain, but broad mid- and hindbrain regions. This contrasts starkly with the presumed plesiomorphic condition in T. paradoxa—which bears short olfactory tracts and a bulbous telencephalic region.

The nasal capsules are large and rounded with a diameter close to 10 mm, and open ventrally (Figs. 3D–3G). Their posteromesial corners open into wide olfactory tracts that are 15 mm long and diverge from each other at 50°. The canals for the orbitonasal vein are large and exit the nasal capsules posterolaterally. Several bony tubules project into the medial rostral space from the olfactory tracts which may have housed the anterior cerebral vein, palatine artery or a ramus of maxillaris nV2.

The telencephalic region is short and low, without any obvious ventral expansion as is common in lungfish (Clement & Ahlberg, 2014). Two large canals for the optic nerves exit the cranial cavity laterally marking the anterior extent of the telencephalic region. In contrast, the diencephalic region is both longer and taller, although of comparable width to the telencephalon. A small dorsal protrusion represents a small pineal eminence, seemingly smaller than those in E. foordi and G. andrewsae. In contrast, the hypophyseal fossa is large. The buccohypophyseal duct opens through a large circular aperture ventrally. Two dorsal-most small canals projecting anteriorly from the hypophyseal region likely housed the ophthalmic arteries, in line with but medial to those is a single small canal that might have housed the internal carotid artery. Below this, two slightly larger canals would have carried the palatine arteries. On the left side of the QMF 21083 on the posterior half of the hypophyseal region is a single small canal that could have carried the pituitary vein.

The mesencephalic region of the endocast is considerably broader than the preceding forebrain (Figs. 3A, 3B). Midway up on the midbrain wall is a canal for the trigeminal nerve (n.V), likely housing both the maxillary and mandibular branches. The rhombencephalic region is very slightly wider than the midbrain, broadening gently towards the labyrinths as in Megalichthys, and in contrast to G. andrewsae which is reconstructed as being narrower in this area. Canals for the vagus nerve (n.X) are visible exiting the cranial cavity posteriorly. Two oval-shaped eminences on the dorsal part of the myelencephalon represent the supraoptic cavities, and the cranial cavity extends a further 14 mm towards and into the intracranial joint.

Although the specimen has been flattened and undergone some crushing, the origin point of the posterior semicircular canals can be identified (Fig. 3A), so together with the lateral extent of the labyrinth, a rough outline of the vestibular system can be inferred. It is not known how large any utricular recess might have been, but the saccular pouches form tear-drop-shaped outlines in ventral view, with rounded anterior margins tapering posteriorly.

The notochordal canal is broad and probably extended as far forward as to be level with the midbrain, although the bounding bone is not preserved well anteriorly (Figs. 3B, 3C).

Hyoid and branchial skeleton

Most of the elements of the hyoid arch and branchial skeleton are preserved in situ within the holotype and can now be described. The hyomandibular and supposed “urohyals” were described by Fox et al. (1995, fig. 43a-d) but their identification of their “urohyal” is more likely median fin basal elements based on comparison with the new scan data. The urohyal revealed in our articulated specimen is a much-elongated bone with a very wide anterior articulation surface for meeting the basibranchial (more detail on this below).

One complete right hypohyal (and a partial left hypophyal, not figured) sit anterolateral to the basibranchial (Figs. 4A, 4C, 4D, 4F, 4G). The hypohyal bears a ball-shaped protuberance proximally for articulation with the anterolateral articular facet of the basibranchial (Figs. 4C, 4D). The distal portion is broadly flared and its shape is similar to those in other tetrapodomorphs, such as Tiktaalik roseae (Downs et al., 2008) and Eusthenopteron foordi (Jarvik, 1954; Jarvik, 1980). It bears a strong ridge running proximodistally along its dorsal surface, as also seen in Holoptychius bergmanni (Cloutier & Schultze, 1996).

The right ceratohyal (Fig. 4D) is a large, mostly flat tear-drop shaped bone, with a smoothed curved anterior margin. It is marked by a large notch in its posterolateral corner for ligamentous attachment. Its shape differs somewhat from those in G. andrewsae (Long, Barwick & Campbell, 1997), T. roseae (Downs et al., 2008), and Medoevia lata (Lebedev, 1995), which have more elongate and narrower ceratohyals, instead it is more reminiscent of the broader bones found in Glyptolepis groenlandica (Jarvik, 1972).

The basibranchial (Fig. 4C) is similar to those in T. roseae (Downs et al., 2008), G. andrewsae (Long, Barwick & Campbell, 1997), M. lata (Lebedev, 1995), and Mandageria fairfaxi (Johanson & Ahlberg, 1997). It forms a slightly elongated heptagonal shape in dorsal view (Figs. 4A–4D). The basibranchial is clearly split (in what appears to be a natural margin) into two transverse halves demarcated by mesiolateral angles that separate it into anterior and posterior portions of similar size. The three lateral and posterior margins are scalloped for articulation with the hypobranchials on each side and the hypohyals anterolaterally.

There are four hypobranchials preserved on the right-hand side (Figs. 4C, 4F, 4G). The first articulates with the anterior portion of the basibranchial, while the second and third articulate with the posterior half of the basibranchial. The fourth hypobranchial, considerably smaller than the other three, is preserved in loose articulation with the posterior section of the third hypobranchial, as in common among sarcopterygians. There are small gaps between the bones which might have been cartilage-filled in life, but this is not visible in the scan data. The hypobranchials are more elongate than the stouter bones in T. roseae (Downs et al., 2008) being more similar to those in M. lata (Lebedev, 1995). The first hypobranchial has a broader anterior margin, and its medial margin is curved more strongly, while the second and third have more or less parallel edges and narrower anterior edges. The fourth hypobranchial is about half the size of the preceding three hypobranchials, but with a similar shape to the third hypobranchial with which it articulates.

In ventral view, a sublingual rod and urohyal are preserved underneath the basibranchial in natural articulation (Figs. 4B, 4D). The sublingual rod is an elongate and narrow bone that tapers slightly anteriorly. It is considerably shorter than that in E. foordi (Jarvik, 1980) but of similar length to that in M. lata (Lebedev, 1995). The urohyal is long and rod-like but does not appear to bifurcate posteriorly nor bear a large posterior flange. In this way the urohyal is similar to that in G. andrewsae (Long, Barwick & Campbell, 1997).

Four ceratobranchials are preserved on both left and right sides of the specimen. Three are long and curved measuring about 45 mm in length, but the fourth is highly reduced and lacks a grooved portion. These are currently under more detailed study in another work currently in preparation by the authors.

The general shape of the hyomandibular (Figs. 4C, 4E) is similar to E. foordi (Jarvik, 1980), although it is not so strongly curved in C. narrienense. It appears to have been a completely ossified bone, more similar to that in G. andrewsae (Long, Barwick & Campbell, 1997) and unlike the unfinished one in T. roseae (Downs et al., 2008). Its proximal extremity is double-headed and its distal end contacts the mesial face of a submandibular bone via its opercular process. There is a large opening between the lateral and medial margins of the proximal portion that would have allowed passage of the hyomandibular canal.

Pectoral girdle and fin

The pectoral girdle of Cladarosymblema narrienense was described by Fox et al. (1995) from several partial or broken bones. There was no supracleithrum visible in the scan of the holotype, however both clavicles, cleithra and anocleithra are well-preserved (Figs. 5A, 5F). Previously, the anocleithra were represented by just two fragments but both complete bones are observable from the scan data (Figs. 5E, 5F). The anterior process is about 12 mm in length and sharply pointed on the left bone, but more rounded on the right-hand side. The posterior flange of the anocleithra is smooth and flat, and measures over 30 mm in length.

Figure 5 Micro-CT 3D rendering (116 µm pixel size) of pectoral and axial elements of Cladarosymblema narrienense (QMF 21082).

(A) ventral view, and (B) in dorsal view, showing placement of bones on holotype; pectoral girdle in (C) dorsal view; and (D) ventral view; (E, F) anocleithra in alternate views; and neural arches in (G) lateral; (I) dorsal; (K) ventral view; ring centra in H, lateral; J, dorsal; L, ventral view.

Similarly, Fox et al. (1995) did not have complete cleithra, but these bones can now be described and illustrated (Figs. 5B, 5D). The cleithra are robust bones with a sizeable branchial lamina. Its external surface is roughened with ornament which consists of irregular tubercles. It has a pointed anterior margin for overlap with the clavicles, but a broad and blunt posterior margin. Although there are some cracks present through this region of the specimen, the cleithrum was a single bone, contra Thomson & Rackoff (1974).

As described by Fox et al. (1995), the clavicles are indeed about half the size of the cleithra and twisted some 40° in orientation. Again, the right-hand side bone is better preserved than the left. Its ventral edge is smoothly rounded, while the dorsal surface bears a thickening. The clavicle bears a long ascending process for articulation with the cleithrum. In addition, there is a small, ovoid bone sitting dorsally above the intersection of the clavicles, interpreted as an unornamented interclavicle, the first time this bone has been identified in this taxon (Fig. 5C).

Fox et al. (1995) stated that the scapulocoracoid and its attachment area are larger in C. narrienense than in other osteolepiforms, but in the holotype is not as extensive as the specimen described and illustrated by those authors. Conversely, the scapulocoracoid in fact appears to be smaller and protruding less than those in Eusthenopteron foordi (Jarvik, 1980), Megalichthys spp. (Andrews & Westoll, 1970a), and Medoevia lata (Lebedev, 1995).

In their original preparation and description of the holotype, Fox et al. (1995) attempted to excavate the appendicular skeleton of the left fin but did not find it, concluding that it must have been poorly ossified. Their radiographs show an outline of some large metaptyerygial elements of the pectoral fin, but artefacts from the metallic wire supporting the perimeter of the fin limit our potential to find any evidence of any ossified pectoral mesomeres in the remaining pectoral fin area of the holotype. In any case, it can be surmised that these may have even been cartilaginous in C. narrienense.

Axial skeleton

As Fox et al. (1995) described in the original description, vertebral ring centra are preserved that are about 11–12 mm in notochordal diameter, and open dorsally (Figs. 5H, 5J, 5L). However, those authors fail to figure or describe the neural arches. The neural arches (Figs. 5G, 5I, 5K) are similar to those figured for Eusthenopteron foordi (Andrews & Westoll, 1970a) and as is common, usually found slightly disarticulated from their associated ring centra (Figs. 5H, 5J, 5L). The two halves of the neural arch join dorsally to form a neural spine, and these are angled about 35 degrees posteriorly from the vertical plane. The neural arches are flat bones with only a very slight tapering at their dorsal tips.

The elements originally identified by Fox et al. (1995) as two isolated urohyals (QMF 26574 and QMF 26573) are reidentified as a proximal basal plate, perhaps from the second dorsal or anal fin (Fox et al., 1995, fig. 43). The proximal end is narrow, whereas the distal end is approximately three times longer showing three articular facets for the distal radials. The morphology of the proximal basal plate of C. narrienense is fairly similar to that observed in E. foordi (Andrews and Westoll 1970, text-fig. 25, 26, 28).

Phylogenetic results

Interrelationships among nine species of megalichthyids and the phylogenetic position of megalichthyids among tetrapodomorphs were analysed using a modified version of the tetrapodomorph matrix used by Cloutier et al. (2020). From the original Cladarosymblema narrienense coding in the Cloutier et al. (2020) phylogenetic matrix, an additional ten characters (72–74, 110–112, 133, 148, 190, 197) were coded based on our new anatomical study. The parsimony analysis (heuristic search) of the complete data matrix (49 taxa, 206 characters) gave 24 948 equally parsimonious trees at 494 steps [consistency index = 0.484, retention index = 0.759], the results from the 50% majority-rule tree is shown in Fig. 6. The general tetrapodomorph topology is similar to the one obtained by Cloutier et al. (2020). The monophyly of megalichthyids has been recovered in the three consensus trees (strict, Adams, 50% majority). Megalichthyids are considered the sister-group to canowindrids. “Osteolepidids” (represented by Osteolepis, Gyroptychius, Medoevia, and Gogonasus) form a grade leading to the clade megalichthyids + canowindrids; the inclusion of additional megalichthyids in our analysis regrouped “osteolepidids” at the base of the clade megalichthyids + canowindrids.

Figure 6 Parsimony analyses.

(A) 50% majority-rule consensus tree from parsimony analysis with inclusion of all taxa showing monophyly of the Megalichthyidae; (B) Canowindrid + Megalichthyid sub-set with most incomplete taxa excluded (Mahalalepis, M. mullisoni, M. laticeps, Sengoerichthys) provides greater resolution of megalichthyid phylogeny. Image silhouettes are authors own (elpistostegalid, rhizodont, megalichthyid, canowindrid) or from PhyloPic, http://phylopic.org/ (Ichthyostega Image credit: Scott Hartman; Eusthenopteron Image credit: Steven Coombs (vectorized by T. Michael Keesey), Gogonasus Photo credit: Nobu Tamura (vectorized by T. Michael Keesey), CC-SA 3.0, http://creativecommons.org/licenses/by-sa/3.0/.

Palatinichthys laticeps is the sister-group of the remaining megalichthyids in the three consensus trees, followed by Megalichthys hibberti. In the 50% majority (Fig. 6A) and strict consensus trees Cladarosymblema forms part of a polytomy including Sengoerichthys, Megalichthyis laticeps and M. mullisoni and [Askerichthys + Mahalalepis]. The Adams consensus tree suggests that two of the most incomplete megalichthyids are responsible for the internal polytomies: Sengoerichthys ottoman (164 unscored and 3 illogical), and Megalichthys laticeps (161 unscored and 6 illogical).

The deletion of the most incomplete megalichthyids [i.e., species with more than 40% of unscored characters: Mahalalepis resima (173 unscored and 0 illogical), Palatinichthys laticeps (135 unscored and 5 illogical), Askerichthys heintzi (130 unscored and 4 illogical), and Megalichthys mullisoni (92 unscored and 6 illogical)] did not modify the position of the remaining megalichthyids on the tree but reduced considerably both the number of steps (471) and the number of equally parsimonious trees (702). An analysis on 44 taxa including a subset of megalichthyids using the best-known species of Megalichthys (i.e., M. hibberti) and excluding the two most incomplete megalichthyids (i.e., Sengoerichthys ottoman and Mahalalepis resima) provide better resolved megalichthyid interrelationships of 481 steps and 648 equally parsimonious trees (Fig. 6B). The consensus trees recovered the following topology: [Palatinichthys [Megalichthys [Askerichthys, Ectosteorhachis, Cladarosymblema ]]].

The monophyly of the megalichthyids is supported by the presence of a long medioventral process of the premaxilla (char. 203), the antero-posterior relationships between the lateral rostral and the anterior tectal relative to the external nostril (character 5; this character could also be phrased as the vertical suture between these two dermal bones at the level of the external nostril). The presence of the anterolateral process of the supratemporal (char. 206) is also shared by most megalichthyids with the exception of Palatinichthys. However, this process is also known in Eusthenopteron, Gyroptychius and Kenichthys. The absence of a pineal foramen (char. 21) characterizes the megalichthyids but is also absent in most of our outgroups.

Discussion

Systematic implications

Cladarosymblema narrienense is significant as the only megalichthyid taxon described from Australia, and along with Mahalalepis resima, one of only two known from Gondwana (Young, Long & Ritchie, 1992). Previously unseen morphological details of the cranial endocast, palate, hyoid and branchial skeleton, pectoral girdle and axial skeleton of C. narrienense are now elucidated, with additional features confirmed or updated from Fox et al.’s (1995) description. These new data, with additional codings from M. resima, and the inclusion of nine megalichthyid species in the parsimony analysis, enabled megalichthyid interrelationships to be reanalysed, with the monophyly of the family confirmed. A full lateral reconstruction of the head of C. narrienense is shown in Fig. 7, as well as 3D renderings of all segmented bones from the holotype (Figs. 7B–7E).

Figure 7 C. ladarosymblema narrienense.

(A) Lateral head reconstruction of Cladarosymblema narrienense, compiled from Fox et al. (1995) and new data. Colour-coded as follows: dermal skull roof (dark blue), cheek (light blue), lower jaw (pale green), opercular series (purple), and pectoral (dark green). Bones marked with “?” remain unknown in this taxon. (B–E) micro-CT 3D rendering of all segmented bones in the holotype (QMF 21082): B, dorsal view; (C) ventral view; (D) anterolaterodorsal view; and E, posteroventrolateral view.

With respect to the phylogenetic status of the Megalichthyidae, Schultze (1974) first identified the specificity of megalichthyids based on the development of the external nares as slit-like openings, partially enclosed by a small posterior tectal bone, and the presence of an intermaxillary process with teeth on the premaxillae.

Seven characters were originally used by Young, Long & Ritchie (1992) to diagnose the Megalichthyidae: (Y+1) elongate or slit-like external naris; (Y+2) partly enclosed by a posterior tectal bone; (Y+3) presence of an intermaxillary process with teeth on the premaxilla; (Y+4) short and broad vomers with a strong mesial process; (Y+5) closed pineal foramen; (Y+6) parietals (“frontal bones“) notched for the posterior nasals; and (Y+7) well-developed lacrimal notch.

Later using an additional 15 features, Fox et al. (1995) provided a general diagnosis for the Megalichthyidae that was not intended to be a phylogenetic diagnosis, but rather a general differentiation from other “osteolepiforms”. Among Fox et al.’s (1995) features that had not been listed by Young, Long & Ritchie (1992), new potential synapomorphies were listed: (F+1) separate bones dorsal and ventral of the narial opening; (F+2) two suboperculars both abutting the posterior-most submandibulars; (F+3) posterior endocranial wall of trigeminofacialis chamber approximately transverse; and (F+4) strong symphysial tusk on dentary and teeth reduced or absent in front of it.

Then, while assessing the phylogenetic position of Litoptychius, Coates & Friedman (2010) mentioned that it shares synapomorphies with megalichthyids including some new neurocranial features: (C&F1) ethmoid articulation for palatoquadrate extends anterior to postnasal wall; (C&F2) nerves II and III exit through common foramen; (C&F3) posteriorly extensive basicranial fenestra; (C&F4) otico-occipital fissure absent; and (C&F5) articular surface of quadrate located above ventral margin of the palatoquadrate.

Next, in their exhaustive study on “osteolepiforms”, Borgen & Nakrem (2016) reviewed features previously used to diagnose the Megalichthyidae to select 11 of which they identified either as indicative, necessary or sufficient to diagnose the group. Among the unambiguous necessary and sufficient characters, they listed five of their 11 characters (we retain their numbering of characters here): (B&N1) anterior palatal dental morphology with the presence of anterior premaxillary tusks (in row or posterior to small, same size marginal premaxillary teeth; their “morphotype C and D”, respectively) in combination with a cosmine covered surface of the cranium; (B&N4) the presence of a branch from the supraorbital sensory canal running towards the anterior tectal (their “postnarial”); (B&N6) a distinct cosmine-less anterior supratemporal (their “intertemporal”) process situated mesial to the opening of the sensory canal (i.e., the supraorbital canal); and (B&N10) a posterior contact between the second submandibular and subopercular and first submandibular.

Most recently, based on the revision provided by Borgen & Nakrem (2016), Downs & Daeschler (2020) reduced the diagnosis to just four synapomorphies: (D&D1) presence of a cosmine cover on dermal bones; (D&D2) a premaxillary tusk that interrupts or lies lingual to the premaxillary marginal tooth row (from Young, Long & Ritchie, 1992); (D&D3) contact between the subopercular and the second submandibular bones (from Fox et al., 1995); and (D&D4) a distinct rostral process of the supratemporal that is without cosmine cover.

Most of these megalichthyid features and synapormophies have already been discussed at length by Fox et al. (1995), Borgen & Nakrem (2016) and Downs & Daeschler (2020). However, additional comments on some of the characters listed above are provided herein. Since Fox et al. (1995), the organization of the bones forming the external naris (F+1) has been recognized as a distinctive feature of megalichthyids. The narial region of megalichthyids in comparison to other “osteolepiforms” necessitates further investigation in order to quantify the size of the naris (Y+1); the relative size of narial surrounding bones (Y+2; char. 204, 205); and the precise trajectory of the sensory canals (B&N4). Although, while megalichthyid external nares seem to be elongated (Y+1), one would have to quantify the shape of the external naris among tetrapodomorphs more broadly to conclusively evaluate this character.

The cheek regions of megalichthyids is poorly known, and although it might reveal some diagnostic features (e.g., shape of the squamosal, shape and height of the dorsal margin of the maxilla, size and orientation of the preopercular) it remains problematic to use these cheek characters diagnostically. The presence of an enlarged anterior tooth on the premaxilla (Y+3 in part, B&N1 in part, D&D2; char. 76, 187) and the presence of a long medioventral process of the premaxilla (char. 203) are present in megalichthyids, but the enlarged anterior tooth of the premaxilla is also present in rhizodonts and some “osteolepidids”.

The absence of the pineal foramen (Y+5; char. 21) was also reported in most of the outgroups used in our analysis; it might well be a plesiomorphic condition or a homoplastic feature among osteichthyans (Janvier, Clément & Cloutier, 2007). Bjerring (1972) suggested that the anterolateral process of the supratemporal (his ”frontodermosphenotic process of the intertemporal” and ”area of intertemporal overlapped by dermosphenotic”) showing a complex articular structure for the parietal and the intertemporal is characteristic of the Megalichthyidae (Janvier, Clément & Cloutier, 2007).

The presence of the anterolateral process of the supratemporal (B&N6 in part, D&D4; char. 206) is shared by most megalichthyids with the exception of Palatinichthys. However, an anterolateral process on the supratemporal is also present in Eusthenopteron, Eusthenodon, Gyroptychius and Kenichthys. The proportion of the vomer (char. 61), as well as the presence of an anteromedial process of the vomer (Y+4 in part; char. 62), should be quantified properly in order to be compared among tetrapodomorphs. The vomers are much broader than long in megalichthyids and some “osteolepidids” and this is also accurate for the presence of the anteromedial process of the vomer.

Concerning the phylogenetic intrarelationships of the Megalichthyidae, taxa previously considered by some researchers (Downs & Daeschler, 2020; Witzmann & Schoch, 2012) to hold dubious affinities (such as Mahalalepis, Palatinichthys and Sengoerichthys) are confirmed as megalichthyid taxa in our analysis.

Previous phylogenetic analyses included three (Ahlberg & Johanson, 1998; Cloutier et al., 2020; Johanson & Ahlberg, 2001; Simões & Pierce, 2021; Zhu & Ahlberg, 2004; Zhu et al., 2017), four (Young, Long & Ritchie, 1992), or five megalichthyids (Witzmann & Schoch, 2012). Thus, our phylogenetic analysis contains the largest megalichthyid diversity included in a phylogenetic analysis with nine species. Megalichthyid intrarelationships recovered from our analysis somewhat resemble that of Witzmann & Schoch (2012) in nesting Palatinichthys laticeps and Ectosteorhachis nitidus close together, and Cladarosymblema narrienense close to Sengoerichthys ottoman, although the position of Megalichthys differs. We propose that this taxon is unstable, influencing the topology whether considered as one taxon or split into species, and is likely paraphyletic. Future analyses will have to include all anatomical features that have been previously discussed in the literature with respect to the phylogenetic status of megalichthyid family, genera and species.

Revised Diagnosis of the Family Megalichthyidae: Tetrapodomorph fishes at a node higher than Osteolepis and lower than Eusthenopteron which have the following characters: small semi-circular shaped lateral rostral and posterior tectal forming the external nostril dorso-ventrally; premaxilla bearing a well-developed posterior process and tusk that interrupts or lies lingual to the premaxillary marginal tooth row; contact between the subopercular and last (or second last) submandibular; and a distinct supratemporal bone with an anterolateral process lacking cosmine cover.

Palaeoneurology

Increasing access to scanning technologies such as synchrotron, neutron and micro- computed tomography (μCT) is advancing palaeontology and in particular the field of “palaeoneurology”, yet still very few tetrapodomorph endocasts are known. Consequently, little is understood about changes to brain morphology during this vital period of evolution approaching the fish-tetrapod transition. In particular, the internal space within the braincase, the “endocast” will prove valuable for developing hypotheses about neural evolution within members on the tetrapodomorph stem. In the absence of preserved brains (which are exceedingly rare), evidence from the extant phylogenetic bracket (lobe-finned fish and amphibians) suggests that it may be possible to make some inferences about the size of certain brain regions from the shape of the endocast alone (Challands, Pardo & Clement, 2020; Clement et al., 2021; Clement et al., 2015).

Megalichthys nitidus (Romer, 1937) and Eusthenopteron foordi (Stensiö, 1963) had their endocasts manually reconstructed in detail via Sollas’ painstaking and destructive grinding method popularized by the Stockholm School (Schultze, 2009). Partial virtual endocasts (ethmosphenoids) have been described more recently of the stem-tetrapod, Tungsenia paradoxa (Lu et al., 2012), and the “osteolepiform” Spodichthys buetleri (Snitting, 2008) from CT data. To date, Gogonasus andrewsae (Holland 2014) remains the only tetrapodomorph for which its full braincase has been investigated via tomographic data, but while the neurocranium was described in detail, a full endocast was neither figured nor described.

With respect to early tetrapod endocasts, a small section of an eroded Ichthyostega stensioei braincase was figured in Clack et al. (2003) illustrating a portion of the oticoccipital, Pardo et al. (2017) figured an endocast from the Early Carboniferous aïstopod, Lethiscus stocki, and the endocast was described from the Permian temnospondyl, Eryops megacephalus (Dempster, 1935).

Thus, the description of the endocast of Cladarosymblema narrienense provides a valuable addition enabling new insight into the neurobiology of the tetrapod stem group. C. narrienense and E. foordi have olfactory tracts shorter and broader than those in M. nitidus and G. andrewsae. The nasal capsules are widely separated from each other and positioned on long olfactory canals, and the forebrain is narrow in all known tetrapodomorphs (except for T. paradoxa), with the mid and hindbrain regions generally being broader and appearing relatively conserved across taxa. While it is problematic to make broad generalisations based on such a small sample, it is striking to note that far greater morphological diversity appears to exist in the endocasts of a comparable group, Palaeozoic lungfish, compared to all known stem tetrapods (Clement et al. Preprint).

The hypophyseal fossa is another region of the braincase that bears further consideration here. The orientation of the hypophyseal region varies among taxa with T. paradoxa (Lu et al., 2012), Diplocercides kayseri (Stensiö, 1963), Youngolepis praecursor (Chang, 1982) and most lungfish (Clement et al., 2016) having small, ventrally-directed hypophyseal fossae. In contrast, E. foordi (Stensiö, 1963), S. buetleri (Snitting, 2008), G. andrewsae (Holland 2014), Qingmenodus jandemarrai (Lu et al., 2016) and several Palaeozoic actinopterygians (Giles & Friedman, 2014) have thinner and narrower ones that extend considerably further ventrally. In C. narrienense and M. nitidus the hypophyseal fossa is a more robust structure, extending ventrally from the cranial cavity but with a sizeable anteriorly-projecting lobe. D. kayseri and Y. praecursor also have significant lobes on their hypophyseal fossae, but these are oriented posteriorly in those taxa.

The anteriorly-oriented space in the hypophyseal region of megalichthyids may potentially accommodate the pars tuberalis as hypothesised for T. paradoxa and Glyptolepis groenlandica (Lu et al., 2012). The pars tuberalis is part of the pituitary gland and is present in all tetrapods but particularly well developed in mammals (Kardong, 2006). It is thought to play a role in sensing photoperiod and was taken as supporting evidence that some brain modifications in stem tetrapods for an increasingly terrestrial lifestyle had appeared as long ago as the Early Devonian (Lu et al., 2012). However, the pars tuberalis, when present, is only a very small upgrowth around the stalk of the infundibulum and may potentially be too small to be reflected in some endocasts. In fact, it is not recognisable in a recent investigation of some extant salamanders (Challands, Pardo & Clement, 2020) nor frog and caecilian endocasts (Clement et al., 2021). This is not to say that animals lacking a pars tuberalis were not sensitive to photoperiod, as even extant fishes which lack a pars tuberalis (chondrichthyans and teleosts) can, for example, sense seasonal changes in day length via their saccus vasculosus instead (Nakane et al., 2013). However, we suggest that the enlarged anterior lobe of the hypophyseal region as seen in C. narrienense most likely accommodates an expanded pars distalis. The pars distalis consists of secretory cells and comprises the bulk of the adenohypophysis (“anterior lobe” of the pituitary) which plays a large role in the production of numerous hormones (Romer & Parsons, 1985).

Conclusions

Synchrotron and µCT of two well-preserved 3D specimens of Cladarosymblema narrienense confirm and update the original description of this taxon, in addition to revealing never-before-seen details of its anatomy, enabling a more comprehensive understanding of the only Australian megalichthyid. This work highlights the value of tomography to supplement traditional preparation and descriptions of key fossil specimens. New details -particularly of the palatoquadrate complex, hyoid and branchial arches, pectoral girdle, and axial skeletons- greatly increase our understanding of this “osteolepidid-grade” tetrapodomorph, boosting our knowledge of the total morphological diversity within this group. In addition, while several cranial endocasts are known from manual reconstructions or isolated ethmosphenoids, C. narrienense enables the reconstruction and visualisation of the first full virtual (from tomographic data) cranial endocast of a tetrapodomorph fish, enabling greater insight into their neurobiological condition, including characteristics of note such as the size and shape of the pituitary gland. A new phylogenetic analysis confirms the monophyly of the Megalichthyidae, which includes seven genera (Askerichthys, Cladarosymblema, Ectosteorhachis, Mahalalepis, Megalichthys, Palatinichthys and Sengoerichthys), and their position within Tetrapodomorpha more broadly. An updated familial diagnosis is provided.

Supplemental Information

Supplemental Information 1 Tomographic data example slice of A,B, holotype QMF 20182 (Micro-CT 3D rendering, scan performed at 58 µm pixel size), and C,D, QMF 21083 (synchrotron-CT rendering, scan performed at 12 µm pixel size); E, X-ray image of holotype QMF 20182 showing total fi

Click here for additional data file.

Supplemental Information 2 Phylogenetic analysis files

Click here for additional data file.

Supplemental Information 3 Animation of tomograms from μCT of Cladarosymblema (QMF 21082) to show quality of scan data

Click here for additional data file.

We are grateful to Scott Hocknull and Kristen Spring (Queensland Museum) for specimen loans. We thank Vincent Dupret (Uppsala University) for assistance with synchrotron scanning and access to beamtime, and acknowledge Flinders Microscopy and Micro Analysis (FMMA) for providing access to the large-volume micro-CT system. We also thank two reviewers, Ted Daeschler and Jorge Mondéjar, whose feedback improved the final version of this manuscript.

Additional Information and Declarations

Competing Interests

Author Contributions

Data Availability

The authors declare there are no competing interests.

Alice M. Clement conceived and designed the experiments, performed the experiments, analyzed the data, prepared figures and/or tables, authored or reviewed drafts of the paper, and approved the final draft.

Richard Cloutier performed the experiments, analyzed the data, prepared figures and/or tables, authored or reviewed drafts of the paper, and approved the final draft.

Jing Lu analyzed the data, authored or reviewed drafts of the paper, and approved the final draft.

Egon Perilli performed the experiments, prepared figures and/or tables, authored or reviewed drafts of the paper, performed scans, processed and modified scan data, and approved the final draft.

Anton Maksimenko performed the experiments, prepared figures and/or tables, performed scans, processed and modified scan data, and approved the final draft.

John Long conceived and designed the experiments, analyzed the data, prepared figures and/or tables, authored or reviewed drafts of the paper, and approved the final draft.

The following information was supplied regarding data availability:

The phylogenetic matrix is available at MorphoBank: http://morphobank.org/permalink/?P3818.

Reconstructed scan data and associated derived files (e.g., STLs and videos) are available at MorphoSource:

https://www.morphosource.org/projects/000383372?utf8=

- https://doi.org/10.17602/M2/M397506

- https://doi.org/10.17602/M2/M397505

- https://doi.org/10.17602/M2/M397501

- https://doi.org/10.17602/M2/M397468

- https://doi.org/10.17602/M2/M397465

- https://doi.org/10.17602/M2/M397462

- https://doi.org/10.17602/M2/M383362

- https://doi.org/10.17602/M2/M397510

- https://doi.org/10.17602/M2/M397514

- https://doi.org/10.17602/M2/M397529

- https://doi.org/10.17602/M2/M397526

- https://doi.org/10.17602/M2/M397523

- https://doi.org/10.17602/M2/M397520

- https://doi.org/10.17602/M2/M397517

- https://doi.org/10.17602/M2/M383374

- https://doi.org/10.17602/M2/M387268

- https://doi.org/10.17602/M2/M383370

- https://doi.org/10.17602/M2/M387265

- https://doi.org/10.17602/M2/M383367

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
