# Peer review of "A fresh look at Cladarosymblema narrienense, a tetrapodomorph fish (Sarcopterygii: Megalichthyidae) from the Carboniferous of Australia, illuminated via X-ray tomography"

_PeerJ, doi:10.7717/peerj.12597_

## Round 0.1 · original submission · Minor Revisions

Two reviewers have given constructive critiques of the manuscript and these generally involve text changes. It seems that the data matrix etc. are on Morphobank so that is acceptable. I agree that the figures are gorgeous. The manuscript is a fine addition to the literature and re-review may not be necessary; however, you need to provide a Track Changes manuscript with your revised submission to aid in that decision. Thank you!

·

Basic reporting

Although the content of this manuscript is generally presented clearly, there are a number of improvements needed:

Starting with the title (and throughout), I don't find the use of the term "tetrapod-like fish" appropriate or helpful. It is not a phylogenetically-precise term (which is particularly needed in the title). I would suggest use of the term stem tetrapod or finned tetrapodomorph instead. The title might be structured more traditionally with the genus/species followed by "(Megalichthyidae; Tetrapodomorpha).

I think the title also needs to include the entire binomial for the taxon under discussion. I think it future-proofs the publication for a day when additional species of Cladarosymblema are in the literature. This same comments applies throughout the manuscript and I would advocate for use of the species name throughout (and especially in figure captions). The use of C. narrienense is certainly a good option when the genus/species has been used previously in a section.

I find a few places where the language seems too casual or unnecessary. Eg. Line 72 - "handful"; Line 101 - "until now"; Line 115/122 - "is a Quennsland Museum specimen"; Line 212 - "managed to piece together". Perhaps these are personal preferences, but I feel like there are too many "herein" and in the pectoral girdle and fin section the use of "we" is unnecessary.

Numerous figure citations are wrong or incomplete, and generally the citations are unevenly used among the different sections of the manuscript. The authors need to review current citations and add additional citations to better help the readers go from text to figures.

I would comment that there are distinct writing-style differences among different sections of the text. The manuscript would benefit from a more consistent style. I'm not suggesting a re-write, only that one of the authors should review the manuscript with this issue in mind.

Line items:
Line 77 - Megalichthyidae also from Late Devonian (Famennian) of North America (M. mullion from Pennsylvania).
Line 138 - Is dataset storage size relevant?
Line 409 - "step" to "steps"
Line 436 - "Stem-tetrapodomorph" to "Tetrapodomorph"
Line 492 - "represent" to "reveal"
Line 560 - Is "under review" appropriate to cite? (even though authors are on both papers)
Figure 7 - Is the dorsoventral shading necessary? I think it takes away from the scientific clarity. Also, is the color-coding of element names in regions necessary? It creates challenges for readers with different color-perception, and when color printers do not perform properly. For example, I do not see any difference between the skull roof (dark blue) and the opercular series (purple).

Experimental design

The imaging and phylogenetic methods used in the manuscript are state-of-the-art. The manuscript provides new data and appropriate interpretation.

Validity of the findings

The anatomical descriptions are well-supported by the new imaging efforts. The phylogenetic methods and hypothesis are adequately presented. The methodological details of the phylogenetic analysis are presented but I do not see any indication that the character list and data matrix will be available.

Additional comments

I enjoyed this manuscript and think it is a very useful contribution, although it does need to be edited/revised.

·

Basic reporting

The study by Clement et al. on the anatomy of the tetrapodomorph fish Cladarosymblema is an important contribution to our knowledge of the diversity and evolution of sarcopterygians.
The text is well written by native English speakers and only a few minor orthographic corrections are needed.
The list of references is extensive and pertinent, although a few additions may be welcome and some errors of citing need to be corrected.
The data are well structured, the figures provide all the necessary information and illustrate well the descriptions in the text, although certain improvements have been suggested (see annotated pdf).
Raw data have been shared and are accessible to interested readers.

Experimental design

The research fits within the aims and scope of PeerJ.
The research questions are well defined, as an exploratory study of the hidden anatomy of previously know and described fossil specimens of an interesting taxon. The three-dimensional preservation of the specimens and the possibilities that virtual microtomography now offers to palaeontologists absolutely justify the use of these techniques in the study of these important fossils.
The methods are correctly described and in detail.

Validity of the findings

The anatomical descriptions use up-to-date terminology and are clearly written, with enough in-depth information.
The conclusions are well stated and reprise the main discoveries of the study.

Additional comments

I have listed a series of comments directly on the pdf file, and I refer the authors to these.

I have, however, certain comments that will need further development by the authors and will be detailed below:

I think further information is needed concerning the status and importance of “osteolepiforms” in the context of sarcopterygian evolution. The term is almost not used in the text and I suspect that it is replaced by “osteolepidids”, which is incorrect. Many readers may be familiar with Eusthenopteron and I consider that a short review of their implications in the study of the fish-tetrapod transition will complement the introduction nicely.

The importance of cosmine in megalichthyids in particular and in tetrapodormorphs in general has not been clearly stated nor explained. The retention of cosmine beyond the Devonian is one of the main features of megalichthyids and may have certainly played a role in the evolution. Many readers may not be familiar with this particular “tissue” and thus a short description and a few extras references may improve the introduction.
I suggest the following ones:
MONDÉJAR-FERNÁNDEZ, J. 2018. On cosmine: its origins, biology and implications for sarcopterygian interrelationships. Cybium, 42, 41–65.
THOMSON K.S., 1975. On the biology of cosmine. Bull. Peabody Mus. Nat. Hist., 40: 1-57.

In general, but especially in the discussion, further information on the paleoenvironment may add more context to the anatomical and evolutionary implications highlighted by the new data. The fact that megalichthyids are primarily freshwater fishes, in contrast with many other early Paleozoic sarcopterygians which are considered as mainly marine fishes, is interesting and should be further developed. Can paleoneurological features be related to ethology or adaptations to certain habitats by megalichthyids? It may be too soon to settle this issue, but some introductory theories may complement the paper and broaden its scope.

Throughout the text and as a piece of general advice, I would initially list the complete name (genus and species) of every taxon, even though in the course of the manuscript only the genus name is used (e.g., Cladarosymblema narrienense first mention, Cladarosymblema after). In case the genus is known by a single species, citing the species every time becomes redundant, but not if there are other species in the same genus.

---

## Round 0.2 · accepted · Accept

Thank you for those meticulous revisions to the manuscript. I have checked them now and am thoroughly convinced that the manuscript has improved markedly and no further review or revision is necessary. Congratulations!!